# Post-Traumatic Stress Disorder in Unaccompanied Refugee Minors: Prevalence, Contributing and Protective Factors, and Effective Interventions: A Scoping Review

**DOI:** 10.3390/children10060941

**Published:** 2023-05-26

**Authors:** Charles Oberg, Hayley Sharma

**Affiliations:** 1Global Pediatrics Program, Department of Pediatrics, School of Medicine, University of Minnesota, Minneapolis, MN 55455, USA; 2Department of Internal Medicine and Pediatrics, Indiana University School of Medicine, Indianapolis, IN 46202, USA

**Keywords:** unaccompanied refugee minor (URM), post-traumatic stress disorder (PTSD), cumulative stress, resilience

## Abstract

In 2021, there were close to 37 million children displaced worldwide. There were 13.7 million refugees and an additional 22.8 million internally displaced. In Europe, this included 23,255 unaccompanied minors seeking asylum, up 72% compared with 2020 (13,550). The objective was to review the current literature regarding PTSD in unaccompanied refugee minors (URM). The authors searched Ovid Medline, Embase, and Cochrane Library from 1 January 2008 through 15 January 2019. Thirty full texts were chosen that specifically studied unaccompanied refugee minors (URM). The results showed that URM had a prevalence of post-traumatic stress disorder (PTSD of 17–85% across the studies reviewed. There were numerous factors that contributed to PTSD, including cumulative stress and trauma, guilt, shame, and uncertainty about legal status. Protective factors included resilience, a trusted mentor, belonging to a social network, religion, having an adult mentor, and having a family (even if far away). Immigrant youth can thrive most easily in multiculturally affirming countries. Five interventions demonstrated effectiveness, comprising trauma-focused cognitive behavioral therapy (TF-CBT); “Mein Weg”, a TF-CBT combined with a group-processing mixed therapy approach; teaching recovery techniques (TRT), narrative exposure therapy for children (KIDNET), and expressive arts intervention (EXIT). The significant mental health conditions include depression, anxiety, internalizing and externalizing behaviors, and frequently PTSD. It is fair to conclude that the high levels of mental health problems experienced in URM are due to exposure to traumatic experiences, separation from parents, and lack of social support.

## 1. Introduction

Violence against children in all forms is a violation of children’s rights and an enormous child health issue [1]. The magnitude and impact of forced migration among children is one such example of violence, associated with physical, sexual, and psychological trauma. The number of children on the move has grown at an unprecedented rate. In 2021, there were 37 million displaced children worldwide including 14 million refugees and 23 million internally displaced children [2]. In Europe, this included 23,255 unaccompanied minors seeking asylum, which is a 72% increase compared with 2020 (13,550) [3]. Most recently, there has been major dislocation of children from Ukraine and Afghanistan. The 2021 withdrawal of U.S. forces from Afghanistan has resulted in Afghan families choosing to separate to get as many members of their families as possible to safety. More than 1500 children have come to the United States unaccompanied [4]. In addition, following the Russian invasion of Ukraine in 2022, 1.1 million children have migrated to neighboring European countries, with an additional 2.5 million internally displaced inside Ukraine [5]. In addition, thousands of Ukrainian child have been forcibly deported to Russia for illegal adoption for Russian families [6].

Post-traumatic stress disorder (PTSD) is prevalent among refugee minors, and its negative effects on daily life have been thoroughly studied [7]. Though there have been studies that have focused on PTSD in unaccompanied refugee minors (URM), they have frequently been cross-sectional, small, qualitative in nature, and/or have used convenience samples.

A URM is defined as “a person who is under the age of 18 and who is separated from both parents and is not being cared for by an adult who by law or custom has the responsibility to do so” [8]. Displacement in all forms puts a child at a high risk of separation from parents. In 2019, only 27% of children surveyed upon arrival in Europe via the Mediterranean had traveled with family [9]. Loss of connection to a caregiver is a developmental crisis for many children amid adversity. Furthermore, in 2017, over 94 refugees arriving in Europe reported indicators of potential trafficking and exploitation during their journey, with children’s reports of these incidents higher than adults [10]. Given how essential a stable family is for a child, it is not surprising that PTSD rates are higher in children who have undergone displacement than in those exposed to similar traumas without the loss of parents and a stable social world [11]. Given the increased instability of unaccompanied children and the subsequent mental health problems that occur, this paper aims to review the literature regarding PTSD in unaccompanied refugee and asylum-seeking minors, focusing on the prevalence of PTSD, protective and exacerbating factors of developing PTSD, and potentially effective interventions.

Even when an unaccompanied refugee minor is able to safely relocate to a new country, the resettlement process may expose them to adverse outcomes of racism, islamophobia, and xenophobia [12]. The purpose of this paper is to provide a review of the prevalence and precursors of PTSD in unaccompanied refugee children, contributing and protective factors, and to provide an overview of potentially effective interventions.

## 2. Methods

The scoping review began with a search of Ovid Medline, Embase, and the Cochrane Library with the inclusion dates of 1 January 2008 through 15 January 2019. In total, 3687 abstracts were screened by 2 independent reviewers. Inclusion factors included unaccompanied asylum-seeking/refugee youth ages 0–18 or PTSD or “post-traumatic stress” mentioned in the abstract. In addition, studies were included if written in English and included both quantitative and/or qualitative studies. Exclusion factors were persons over the age of 18 years, former child refugees, and studies reported in languages other than English. If the abstracts mentioned PTSD in reference to unaccompanied refugee and/or asylum-seeking minors, they were included in a secondary analysis which generated 30 full-text articles included in the review and analysis. The Appendix A contains a list and description of the original 30 studies. A subsequent search was undertaken on 30 April 2023, for the years 2019–2022 to identify additional relevant research to augment the initial scoping review. A limitation in this review design was that it was limited to articles in English, resulting in a body of work in other languages that was not examined.

## 3. Findings

### 3.1. Countries of Origin and Resettlement

Figure 1 provides a depiction of both the original home countries of origin identified for the unaccompanied refugee minors (URM) and the countries of resettlement.

In the studies reviewed, the URM came from 51 different countries of origin from three global regions. Sub-Saharan Africa accounted for 25 countries of origin and the Middle East and North Africa accounted for 15 countries. The remaining refugees came from 13 countries from the European Balkans and Asia. The breakdown of countries by regions is as follows:***Sub-Saharan Africa:*** Somalia, Sudan, Nigeria, Ghana, Gambia, Eritrea, Congo, Niger, Mali, Kenya, Ethiopia, Guinea-Bissau, Guinea, Senegal, Angola, Sierra Leone, Algeria, Liberia, Uganda, Rwanda, Burundi, Senegal, and Tanzania.***Middle East and North Africa:*** Afghanistan, Turkey, Libya, Morocco, Algeria, Tunisia, Egypt, Yemen, Syria, Iraq, Iran, West Bank & Gaza, Lebanon, Azerbaijan, and Saudi Arabia.***Asia:*** China, Sri Lanka, Bangladesh, Philippines, Burma, Nepal, Bhutan, and Pakistan.***Europe/Balkans:*** Albania, Serbia, Kosovo, Macedonia, and Moldova.

In the studies reviewed, URM resettled in 11 countries comprising the United Kingdom, Germany, Norway, Sweden, Netherlands, Denmark, Belgium, Austria, Italy, Australia, and the United States.

### 3.2. PTSD Prevalence and Mental Health Comorbidities

Out of the 30 studies reviewed, there were 15 that attempted to specifically determine the prevalence of post-traumatic stress disorder among URM. Table 1 provides a summary of the 15 studies identifying the 8 different tools used to determine the PTSD prevalence.

The prevalence of PTSD ranged from 17% to 85% with a mean prevalence rate of 46%. Michelson and Sclare reported the highest PTSD prevalence rate of 85% for unaccompanied youth compared to 66% among accompanied minors [13]. However, the researchers did not specify the measurement tool used in their study. 

The study that had the next highest prevalence rate was conducted by Sarkadi et al. and identified PTSD among the URM 8–18 years old at 76%. They utilized the CRIES-8 tool that consists of a short questionnaire which was used during their routine health check-up [15]. McGregor et al. utilized a measurement tool called the Child PTSD Symptom Scale (CPSS), a 17-item self-report inventory based on DSM-IV criteria. The researchers observed a PTSD rate of 59% for minors who were alone compared to a rate of 30% for those with intact families [17].

A frequently utilized measurement tool is the Reactions of Adolescents to Traumatic Stress (RATS) questionnaire. The RATS is a 22-item, self-report screening tool developed and validated that corresponds with the DSM-IV criteria for PTSD [28]. Stoltz et al. used the UCLA PTSD Index and found that 28.1% of the children and youth exceeded the PTSD threshold. In addition, cumulative stress to traumatic stress posed a risk factor for mental health and functional impairment [29]. Huemer et al. found a PTSD rate of 19.5% using the Mini-International Neuropsychiatric Interview for Children and Adolescents (M.I.N.I. Kid) measurement tool [26]. The last measurement tool utilized in the studies reviewed was the Harvard Trauma Questionnaire (HTQ) used by Geltman et al., who ascertained a PTSD rate of 20% [25]. The studies consistently revealed a higher prevalence of PTSD among unaccompanied minors compared to refugee children traveling with families. 

In addition to PTSD, there are other significant mental health problems that are associated with being an URM, such as anxiety and depression associated with internalizing and externalizing behaviors. Derluyn and colleagues identified that in addition to elevated overall PTSD scores on the RATS, refugee adolescents separated from their parents had significantly higher depression scores based on the Hopkins Symptom Checklist-37 (HSCL-37) scale [30]. Jackoben and colleagues using the HTQ found a PTSD rate of 30.6% and an associated major depression rate of 9.4% and an anxiety disorder rate of 3.9% [22]. Sanchez-Cao revealed that even with high rates of PTSD (66%) and depression (12%) identified in URM, they were frequently not in contact with mental health services [16]. Smid et al. examined for PTSD at both 1 year (T1) and 2 years (T2) after resettlement. The researchers found a 40% prevalence of PTSD at T1 and an additional 16% PTSD at time T2. In refugee minors with late-onset PTSD, PTSD symptoms 2 years following resettlement appeared to be mediated by symptoms of depression and anxiety [20]. Research has recently shown that suicide ideation and suicide are increased in URM compared to the host populations of the same age [31].

### 3.3. Contributing Factors

There were numerous factors that exacerbated PTSD among URM. Cumulative trauma and stress were the single most important contributing factors. Cumulative stress is defined as the total number of stressful and traumatic events as measured by the Stressful Life Events Questionnaire (SLE), which contains 12 yes and no questions concerning events pertaining to loss, conflict, and violence. Bronstein et al. conducted a bivariate analysis that indicated that cumulative stress and trauma were correlated with higher levels of post-traumatic stress (*p* < 0.001) [21]. Vervliet et al. revealed that the mean number of daily stressors continued to increase over time. In addition, there was a high prevalence of PTSD (52.7%), depression (44.1%), and anxiety (38.3%) among children on the move, which are associated with the number of traumatic events the minors reported [19]. 

Jensen et al. found stressful life events actually increased over time from initial relocation to several years following their migration [18]. Mueller-Bamouh and colleagues explored the impact of exposure to violence on PTSD symptoms and found that PTSD severity was correlated with the exposure to familial and organized violence [32].

Feelings of guilt and shame are associated with the number of traumatic events that URM experience. Geltman and colleagues, using the Trauma-related Guilt Inventory and Shame Variability Questionnaire, found that post-traumatic guilt and shame in URMs were significantly correlated with both PTSD symptoms and severity. The cumulative stress of exposure to multiple traumatic events poses a significant risk factor for mental health including greater suffering and functional impairment due to shame and guilt [24]. Verliet et al. examined the risk of developing later post-traumatic stress disorder (late PTSD) [33]. This study is unique in that it is a longitudinal study. The researchers followed up on the psychological wellbeing of a sizeable group of unaccompanied minors from arrival in the host country to one and a half years later, using a linear mixed model approach. What is disconcerting is that the study’s results indicated that mental health problems in URM often persist for a significant period of time. Certain factors were identified that increased the levels of PTSD symptoms in refugee minors, including experiencing violent death of a family member, being unaccompanied, and severity of the exposure to stressful life events. 

Other contributing factors included female gender and trauma, increasing age, less education, and refusal of asylum. Most of the unaccompanied refugees had experienced life-threatening events, physical abuse, or loss of a close relative. The majority of the papers included in the review were primarily of male URM. In fact, across the 30 studies, a total of 4690 URM were studied with 88% of the participants male and just 12% female participants. Yet, in Volkl-Kernstock et al., gender was a major contributing risk factor with females having a higher prevalence of PTSD [27]. 

Hodges et al. demonstrated that PTSD was significantly higher among those in low support living arrangements, i.e., living independently or semi-independently [34]. Bronstein and Montgomery conducted a study observing sleep patterns among unaccompanied Afghan refugees and their association with PTSD. PTSD was associated with later bedtimes, less total sleep, and an increase in nightmares [35]. The mental health trajectory of young asylum seekers appears to be negatively affected by low support during the asylum process and uncertainty or refusal of asylum [36]. 

### 3.4. Protective Factors

The resilience of URM is a major protective factor in reducing the risk of PTSD. Resilience refers to the capacity of a complex adaptive system (including families, communities, economies, and other kinds of systems) to respond effectively to challenges that threaten the survival, function, or development of the system. Historically, resilience science emerged from research designed to understand the etiology of psychopathology by studying children with well-known risk factors associated with problems in development [37]. Pioneering investigations realized that some children fared adequately well or even thrived in the aftermath of adverse experiences [38]. Subsequent research has aimed to account for the differential life trajectories observed in the aftermath of risk exposure, including political conflict, war, and displacement [39]. Such research has played a significant role in highlighting the striking variation in children’s post-traumatic experiences [40]. 

Resilience can be defined as the attainment of desirable social outcomes and emotional adjustment despite exposure to considerable risk, and it reflects a child’s positive adaptation in the context of significant adversity [41]. A resilient child is able to meet developmental milestones and enjoy competence despite exposure to adverse circumstances [42]. Thus, it is not surprising that family functioning and the network of caring relationships surrounding a child play a crucial role in child resilience [43]. The key to awakening the child’s resilience and agency is largely by attempting to identify and create protective factors in both the child and the environment [44]. These include belonging to a social network, religion, or having an adult mentor and maintaining a family connection with family who may be back home or displaced to a different location. The most significant protective factor was the levels of resilience in both pre-migratory and peri-migratory experiences [45]. A secured refugee status also provides significant protection from PTSD and other emotional, behavioral, and mental health conditions. 

Contextual factors have significant impact on the mental health status of refugee and asylum-seeking youth. The importance of a trusting relationship while working with unaccompanied youth has proven to be a significant protective factor in regard to mental health. Maintaining a connection to one’s country of origin affirms the uniqueness of the culture. Different countries have separate systems by which they review asylum claims and provide different rights and entitlements to claimants. Contact with and ongoing social support from friends and family abroad can blunt the effects of discrimination and assist in acculturation and improvement in mental health. Specifically, ongoing support had a significant direct positive impact on the effects of depression and indirect effects by increasing cultural competence [46]. A major predictor for positive mental health outcomes is belonging to a social network and the support during the child’s flight and migration. Jarlby and colleagues observed that perceived social support from family peers and adult mentors can moderate the relationship between stressful life events and the mental health of the unaccompanied refugee minor. However, discussing past traumatic events may be associated with exacerbating mental health problems among young refugees [47]. Sierau et al. found that a high level of perceived mentor support diminished the association between the number of stressful life events with PTSD symptoms, depression, and anxiety [23].

There have been few studies on the post-migration educational outcomes of URM. In particular, what factors contributes to educational success? O’Higgins et al. found that living arrangements were significantly related to the school success for URM. Specifically, living in a foster care setting, particularly in same-ethnic foster care, as opposed to a detention facility resulted in significantly better education outcomes [48]. It has also been shown that detention of children upon arrival in their new host country is detrimental to their socioemotional health, resulting in behavioral difficulty [49]. Knowledge about the home, school, and community for URM is essential for promoting the ideal environment for the youth to academically thrive [50]. In addition, access to school-based mental health for asylum and refugee children appears to provide timely and appropriate intervention to a population of young people who do not traditionally access mental-health services [51].

Finally, focusing on intercultural competence, emotional regulation, and goal setting fosters adaptation by giving knowledge about cultural difference in values and norms. Emotional regulation is a process of the person’s capability to evaluate, manage, experience, express, and improve emotional reactions in a way that helps proper functioning. Emotion regulation may assist in the acculturation process and may be helpful to deal with traumatic experiences [52]. Emotional regulation regarding empathy, positive reappraisal, and cultural difference in emotion expression fosters both adaptation and improved mental health [53]. 

## 4. Interventions for PTSD in URM

There are several interventions that have demonstrated to be effective. These include trauma-focused cognitive behavioral therapy (TF-CBT); My Way (“Mein Weg”), a TF-CBT and group-processing mixed therapy; teaching recovery techniques (TRT); expressive arts intervention (EXIT); and narrative exposure therapy for children (KIDNET). Each is briefly described below.

### 4.1. Trauma-Focused Cognitive Behavioral Therapy (TF-CBT)

Trauma-focused cognitive behavioral therapy (TF-CBT) is considered one of the most widely studied and effective treatment of post-traumatic stress disorder for school-aged children and adolescents [54]. Gutermann et al., in a meta-analysis of 135 studies, found that TF-CBT was the most effective and yielded the highest mean effect sizes regarding treatment of those who have experienced trauma [55]. In regard to URM, Uterhitzenberger and Rossner, in a pilot study, found that TF-CBT was feasible and effective in reducing PTSD in severely traumatized in an unaccompanied refugee minor girl [56]. TF-CBT included the following eight components under the PRACTICE acronym: psychoeducation and parenting skills, relaxation, affective modulation, cognitive processing, trauma narrative, in vivo exposure, conjoint child/caregiver sessions, and enhanced safety and future skills. The individual participated in 12 sessions of TF-CBT from a treatment manual, with PTSD symptoms decreasing in a significantly and remaining stable for 6 months. However, since it was a case report of a single individual, one must be cautious on generalizing the results. The same authors conducted a second pilot with six URM with similar positive outcomes [57]. More recently, Chipalo conducted a systematic review of the effectiveness of TF-CBT in reducing trauma symptoms among refugee children and found that in four peer review studies TF-CBT were deemed effective [58]. Despite the promising results of TF-CBT with URM, future additional studies need to be conducted to provide more empirical support for its effectiveness and generalizability to for treatment of PTSD with unaccompanied refugee youth.

### 4.2. My Way or “Mein Weg”

My Way or (“Mein Weg” in German) is a short-term component-based intervention of TF-CBT combined with a group-processing mixed therapy approach. The cognitive behavioral components compromised psychoeducation, cognitive restructuring, and promoting enhanced safety and future development. The meditation practice included relaxation technique and deep breathing. Pheiffer and Goldbeck conducted a pilot study published in 2017 with 29 Afghan male URMs aged 14–18 years. The pre/post-test comparison indicated a reduction in PTSD symptoms. The intervention demonstrated reduced physiological manifestations of stress, startle response, hypervigilance, sleeping, and attention problems, and anger dysregulation [59]. The young refugees reported significantly fewer symptoms after taking part in the intervention compared with before the intervention. Improvement was significant in the domains of re-experiencing and avoidance and also in improvement in cognition and mood. In 2019, Pfeiffer and colleagues conducted a randomized, controlled trial with 50 male participants (randomly assigned to the Mein Weg or usual care (UC) group). Both the initial post-treatment and 3-month follow-up assessments showed that the Mein Weg group interventions were more effective in reducing PTSD and depression symptoms [60].

### 4.3. Teaching Recovery Techniques (TRT)

Teaching recovery techniques (TRT) was developed by The Children and War Foundation, based in the United Kingdom and Norway, and is another manualized intervention based on TF-CBT in a group setting [61]. TRT is a low-threshold, five-session intervention developed to reduce distressing war- and disaster-related trauma reactions among children and youth. It includes both stress-management skills that help children to better process their trauma-related emotions as well gradual exposure to traumatic experience to assist children in gaining mastery over traumatic reminders. TRT enables normalization to trauma and offers children emotional support by providing them with strategies to cope. Sarkadi and colleagues utilized the CRIES-8 and the Montgomery–Asberg Depression Rating Scale Self-report (MADRS-S) at baseline and post-TCT intervention. At baseline, 83% of URMs were reported to be moderately or severely depressed with 48% experiencing suicidal ideation. Both the PTSD and depression symptoms decreased significantly after intervention [14]. Saradi et al. initiated a TRT study to evaluate the effect and efficiency of this approach in a randomized, controlled trial [62]. However, it became evident that revisions to the protocol were needed, especially with regard to the procedures for recruitment and randomization. Upon revision, an adequately powered randomized clinical trial is presently underway [63]. Recently, Solhaug and colleagues demonstrated that TRT is a potential useful intervention to enhance life satisfaction among URM and can be a measure to support positive development among youth at risk for mental health problems [64]. Finally, El-Khani and colleagues combined TRT and a parenting education program with preliminary outcome data showing that TRT plus parenting may have the potential to reduce refugee children’s trauma and increase caregivers’ parenting self-efficacy [65].

### 4.4. Expressive Arts Intervention (EXIT)

DeMott et al. described the use of expressive arts intervention (EXIT) with unaccompanied asylum-seeking children. The approach is based on intermodal expressive arts consisting of music education and music/art/dance therapy to access the “play space”. The objective was to study the long-term effects of short-term, early group intervention. The results demonstrated that EXIT had a beneficial effect on helping minor refugee boys with symptoms of trauma and post-traumatic stress symptoms (PTSS). Improvement was noted in a greater sense of safety, calming, connectedness, self- and community efficacy, and hope. At the end of the follow-up, the EXIT group had a beneficial effect on helping minor refugee boys cope with symptoms of trauma, increase life satisfaction, and develop hope for the future compared to the control group [66]. 

### 4.5. Narrative Exposure Therapy for Children (KIDNET)

KIDNET is a short-term psychotherapeutic approach to the treatment of PTSD for children. It consists of the child and therapist constructing a chronological narrative of the child’s exposure to traumatic stress over the course of 6 to 10 sessions. Ruff et al. demonstrated success in the treatment of refugee children experiencing PTSD symptoms with positive results lasting for over a year [67]. More recently Peltonen and Kangaslampi (2019) randomly assigned 50 severely traumatized children and adolescents, including 37 refugees, in Finland to KIDNET and treatment-as-usual groups (TAU). KIDNET was significantly more effective than the TAU in reducing PTSD symptoms [68]. There is presently a rater-blinded, multi-center, randomized, controlled trial comparing KIDNET to treatment as usual within the German general healthcare system [69].

Finally, it should be noted that there are additional emerging approaches that may prove efficacious with URM, specifically, the extinction of conditioned fear as shown in people suffering from post-traumatic stress disorder (PTSD) and anxiety. Neural oscillatory correlates of such mechanisms appear relevant regarding the development of novel therapeutic approaches such as electrical brain stimulation, vagal nerve stimulation, and deep brain stimulation [70].

## 5. Conclusions

The arc of migration for URM includes the trauma they experience in their home country, the struggles during their time in transit, and finally the challenges experienced in their new countries [71]. The first includes experiencing the violence of armed conflict firsthand with the frequent loss of a loved ones. Second, the URM fled their home and community and experienced trauma during their journey, including physical and sexual abuse, trafficking, and the stress of living in refugee camps. Finally, they live the stresses of the transition to a new home community and country. Even when an unaccompanied minor is able to safely relocate to a new country, the resettlement process may expose them to adverse outcomes of racism, islamophobia, and xenophobia. The plight of these children must be characterized as a group of the most vulnerable children today. The significant mental health conditions include depression, anxiety, internalizing and externalizing behaviors, and frequently PTSD. It is fair to conclude that the high levels of mental health problems in URM are due to separation to their parents, exposure to traumatic experiences, lack of parental and social support, and multiple losses. This is exacerbated by their uncertain future regarding asylum and their current living situation. Increased attention should be given to post-migration risk and resilience factors, especially environmental and relational factors [72].

One of strengths of this scoping review was that the studies reviewed 4690 URM coming from as many as 51 countries of origin and 11 different countries of resettlement. The migration picture once again demonstrated the movement from low- and middle-income countries (LMIC) to high-income countries (HIC). The prevalence of trauma and subsequent identification of emotional and mental health problems were consistent across the globe. Another significant strength is that the paper for the first time identified and reviewed the interventions that have proven to be most effective in the treatment of unaccompanied refugee and asylum-seeking children suffering with PTSD.

A key finding was that a number of different measurement tools were used in the documentation of PTSD and mental health difficulties, and yet, there was limited comparison across these tools. A suggestion for future research would be the use of multiple tools in concert to demonstrate the validity across platforms to potentially identify a gold standard for future investigations. 

A second key finding is greater attention to the impact of gender needs to be explored in future research. As can be seen in the Appendix A, the majority of studies had significantly greater numbers of male (88%) as opposed to female (22%) participants. Yet, as previously stated, female gender, when studied, showed a significant level of PTSD greater risk greater than the male participants. Another area for future research would be greater attention to the impact of age from a developmental perspective. The age range for the studies reviewed was as early as 9 years of age to as late as 21 years of age, with the majority focusing on the later adolescent years from 15–18 years of age. Yet, there are significant changes that occur from late childhood to late adolescence and young adulthood. An exploration of the prevalence of PTSD and other mental health outcomes stratified and analyzed by age could provide insights into severity and contributing and protective factors as impacted by their developmental age. It would also be insightful to explore their age specific responses to adversity and the capacity for resilience. In addition, such efforts would be invaluable in examining the effectiveness of intervention based on the URM age.

Children on the move can thrive most easily in multiculturally affirming countries, schools that are well resourced for supporting and integrating youth from diverse national backgrounds, neighborhoods that foster connection to culturally conferred protective factors, and healthily connected families. Every effort must be made on national and community levels to ensure that children are situated in such networks. It is important to use culturally effective therapeutic approaches that help to heal the wounds of their journeys. Every effort must be undertaken to promote a trauma-informed care (TIC) approach to the care of URM to make sure that they are not retraumatized in the process of our efforts to support, assist, and treat them [73]. Finally, as research continues, it is imperative that longitudinal studies are conducted to monitor the long-term effectiveness of our interventions.

## Figures and Tables

**Figure 1 children-10-00941-f001:**
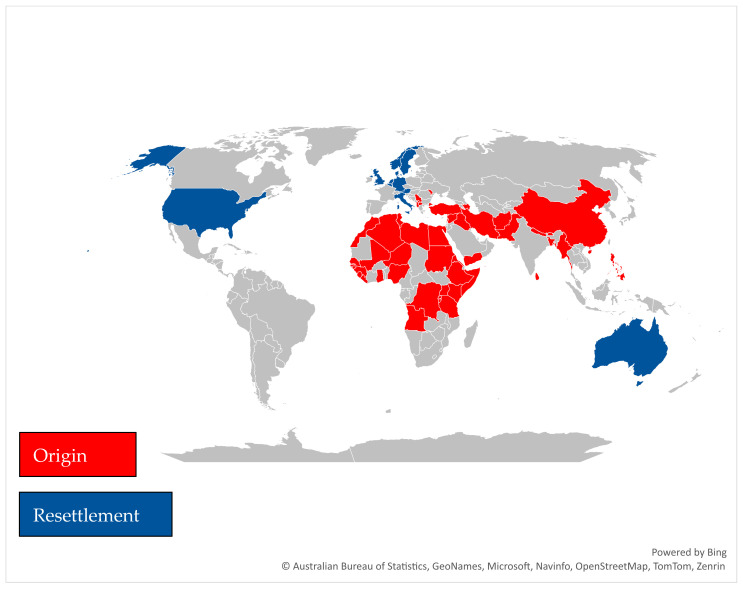
Countries of Origin and Resettlement.

**Table 1 children-10-00941-t001:** Prevalence of PTSD and Scales Utilized.

Authors	Methods/Tool	URM
Michelson and Sclare [13]	Tool not identified	85%
Sarkadi et al. [14]	CRIES-8	76%
Salari et al. [15]	CRIES-8	76%
Sanchez-Cao et al. [16]	HTQ	66%
McGregor et al. [17]	CPSS	59%
Jensen et al. [18]	CPSS	54%
Vervliet et al. [19]	RATS	52.7%
Smid et al. [20]	RATS	40%
Bronstein et al. [21]	RATS	34.3%
Jackoben et al. [22]	HTQ	30.6%
Sierau et al. [23]	PCL-5	30.5%
Stotz et al. [24]	UCLA PTSD Index	28.1%
Geltman et al. [25]	HTQ	20%
Huemer et al. [26]	M.I.N.I. Kid	19.5%
Volkyl-Kernstock et al. [27]	UCLA PTSD Index	17%

RATS—The Reactions of Adolescents to Traumatic Stress. M.I.N.I. Kid—The Mini-International Neuropsychiatric Interview for Children and Adolescents. The UCLA PTSD—Reaction Index for DSM IV (Revision 1). HTQ—the Harvard Trauma Questionnaire. CPSS—The Child PTSD Symptom Scale. CRIES-8 questionnaire—Revised version of the Children’s Revised Impact of Event Scale. PCL-5—the Post-Traumatic Stress Disorder Checklist. CAPS-CA is used to obtain PTSD diagnoses according to the Diagnostic and Statistical Manual of Mental Disorders, 4th edition (DSM-IV).

## Data Availability

Not applicable.

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
