# Peer review of "Post-Traumatic Stress Disorder in Unaccompanied Refugee Minors: Prevalence, Contributing and Protective Factors, and Effective Interventions: A Scoping Review"

_children, 2023, doi:10.3390/children10060941_

Round 1

Reviewer 1 Report

Brief reviewer’s summary : 

The aim of this scoping review is to provide findings on the prevalence of PTSD in unaccompanied refugee children, contributing and protective factors, and finally to provide an overview of potentially effective interventions. The paper provides an interesting overview of these data and is easy to read.

General comments of the reviewer : 

The authors have worked hard to collect the data. The manuscript is clear and easy to read, presented in a well-structured way. The topic is interesting and important. The authors have read many abstracts, but the results are already well known. The references cited are not very recent, they stopped at the beginning of 2019, and many articles were written between 2019 and 2022. I would prefer more new references for an article to be published in 2023. Unfortunately, the article does not advance current knowledge on the subject, as all the things that are explained are well known. The only thing that may be new is the analysis of effective interventions. But there is another problem. In my humble opinion, the last part of the document "Effective interventions" is not well analysed. The references are not appropriate in my view. The authors suggest that there are several interventions that have been shown to be effective, but the references given are not sufficient to prove this, and the criteria for choosing the references are not clear either. I took the time to read the articles in the references. For example:

·       For TF-CBT, the authors provide two references that are only case reports, the first with 6 cases, and the second with one case. Both studies are interesting, but not sufficient. Moreover, they are both from the same author. If that is the criterion, then there are many other approaches that can be considered effective.

·       For the intervention called "My way" is concerned, it is a small pilot study, interesting, but in my opinion not enough to say that it is a "proven intervention". Much more research needs to be done on this method to prove it.

·       For the "Teaching Recovery Techniques", the reference is from 2018. The study is interesting and well done, and the results are consistent, showing that this intervention can be effective, but the study was not sufficient to confirm this. Then the same group of researchers worked on a very interesting protocol, published in 2020. The aim of this second study was to do a randomised trial, but unfortunately the study was not completed due to recruitment problems and finally the Covid pandemic, as they mention in their 2022 publication. I hope they will continue to advance the research in the future. "Rondung.E, Leiler.A, Sarkadi.A, Bjärtå.A, Lampa.E, Löfving.S, Calam.R, Oppedal.B, Keeshin.B, Warner.G. (2022). Feasibility of a randomised trial of Teaching Recovery Techniques (TRT) with refugee youth: results from a pilot of the Swedish UnaccomPanied yOuth Refugee Trial (SUPpORT). Pilot and feasibility studies. 8. 10.1186/s40814-022-00998-1.”

·       The "Expressive Arts Intervention (EXIT)" study is interesting and well done. The intervention can be considered effective in improving the quality of life of unsupported young people.

·       Finally, regarding the "Promotion of acculturation and intercultural competence", there are many empirical studies and some research proving its effectiveness.

But in general, I cannot consider this section as consistent. I think this part needs a better analysis and needs to be completely reanalysed.

Finally, I find some errors in the references section : 

·   Many references hasn’t the right format. The authors must correct that and use the same format for all the references. 

·  N°37. Pag16. The reference it’s not complet. It must be “ Unterhitzenberger J, Eberle-Sejari R, Rassenhofer M, Sukale T, Rosner R, Goldbeck L. Trauma-focused cognitive behavioral therapy with unaccompanied refugee minors: a case series. BMC Psychiatry. 2015; 15:260. doi: 10.1186/s12888-015-0645-0. PMID: 26497391; PMCID: PMC4619299.

· N°39. Pag 17. The title it’s not complete too. “Pfeiffer E and Goldbeck L. Evaluation of a Trauma-Focused GROUP Intervention for unaccompanied Young Refugees-A pilot study. Journal of Traumatic Stress, 2017; 30: 531-536”

I have no specific comments referring to tables or figures that point out inaccuracies within the text or sentences that are unclear. 

Specific comments to the authors :

· Pag. 2, §1, “However, fewer studies have focused on PTSD in unaccompanied refugee minors (URM)”. This affirmation needs to be more precise, because there are many studies about it, in Europe, but may be the limits are that many studies are qualitatives, and not quantitatives ? Or not all in English, but many in French ? I think that this eventual limitation must be explained. 

·   Pag. 2, §2, « Only 28% of children surveyed upon arrival in Europe via the Mediterranean had traveled with family ». Needs a reference.

·  Pag. 3, §2, “Deficit mindset” needs a little bit more of precision about what means. 

·   Pag. 3,§3. The scoping review began in 2008 and ended in January 2019. Why the authors didn’t follow until 2021 or even the beginning of 2022 ? There are quite a lot of studies during the periode 2019 - 2022. 

·  Pag.9,§1: “The most significant protective factor was the levels of resilience in both pre-migratory and peri-migratory experiences. (33) A secured refugee status also provides significant protection from PTSD and other emotional, behavioral, and mental health conditions”. The idea is not clear. Resilience is rarely active at the time of trauma. Resilience is active after the event, usually in the post-migration experience. I think that the concept of resilience, which is very interesting, needs to be better defined. Moreover, many studies show that post-migration factors are as important as, or even more important than, pre- and peri-migration factors.

Author Response

The authors want to thank the reviewer for thoughtful review and critique. The many suggestion have been incorporated into the revision of the paper.  Please see the file below for the complete response.

Reviewer 2 Report

I am pleased to have had the opportunity to review this paper. It is very timely considering 250,000 undocumented minors have crossed the US border over the last 2 years.

Our organization has been asked to provide transitional foster care for those undocumented minors who are waiting in detention centers to be reunified with family. I am anxious to share your paper with our case managers and mental health providers.

My only recommendation is that in the conclusion section it is worth mentioning the more recent influx of minors crossing the borders due to recent conflicts in Afghanistan and Ukraine. I realize the time necessary to conduct research and prepare for publications. But it is possible for the authors to provide a more recent context in the conclusions and future research section. We have become aware that many youth in Ukraine were voluntarily and involuntarily moved to camps in Russian territory and have been traumatized. It is essential that we understand the level of trauma these youth have experienced in order to provide support and interventions.

Thanks for this work. I hope we can build upon it.

Author Response

The authors express their gratitude to the reviewer. 

The authors concur and several sentences regarding Ukraine and Afghan children were added not in the conclusion but to the Introduction. Specially,

Most recently, there has been major dislocation of children from Ukraine and Afghanistan.  The 2021 withdrawal of U.S. forces from Afghanistan has resulted in Afghan families choosing to separate to get as many members of their families as to safety. More than 1500 children have come to the United States unaccompanied.  In addition, following Russians invasion of Ukraine in 2022, 1.1 million children have migrated to neighboring European countries with an additional, 2.5 million internally displaced inside Ukraine. In addition, thousands of Ukrainian child have been forcibly deported to Russian for illegal adoption for Russian families.

Reviewer 3 Report

This work addresses the important issue of PTSD in a vulnerable population, namely the case of unaccompanied refugee minors. I have few observations that require your attention:

-Why the inclusion criteria is people with a range of age 0-25 years, when referring  to minors?  please correct the inaccuracy.

-With regards to novel interventions targeting PTSD, it would be relevant to discuss about non-invasive neuromodulation approaches that focus on extinction of fear conditioning (please see https://doi.org/10.3390/biomedicines6020049)   

Author Response

The authors would like to extend their appreciation to the reviewer for their critique.

The reviewer is correct, and the age was changed to l8 years of age. In addition, in the introduction, a sentence was statingURM is defined as “a person who is under the age of 18 and who is separated from both parents and is not being cared for by an adult who by law or custom has the responsibility to do so”, with new reference.

In addition, the authors found the article on neural oscillatory correlates to be very interesting as a possible approach to the treatment of URM.   A sentence was added at the end of the intervention section highlighting the approach.

Round 2

Reviewer 1 Report

Manuscript’s 2nd Review

12.05.2023

Brief reviewer’s summary : 

The authors have actualized the introduction with interesting new information, allowing the article to be more actual. Also, they explain why they are limited to only studies written in English. But, unfortunately, the main problem for this paper remains the same. As I mentioned in the previous review, the authors suggested that there were several interventions that have been shown to be effective, but the references given were not sufficient to prove this. The authors have work on its part and propose new references. I took the time to read the articles of the references, but unfortunately, my conclusion is still the same. Further, in the “specific comments to the authors”, I give you the arguments. 

Specific comments to the authors :

In the first review I have suggested some corrections, that all have been considered. 

·       Pag. 2, §1 (now 2§2) : now is more clear. 

·       Pag. 2, §2 (now 3§1): good reference. 

·    Pag. 3, §2 (now 3§3) : “Deficit mindset” needs still a little bit more of precision about what means. 

·       Pag. 3,§3 (now 4§1): The scoping review has been enlarged to studies from 2019 to 2022. 

·       Pag.9,§1 (now 10§): the concept of resilience is better defined. 

·     Pag.12 : Concerning the “Effective Interventions”: As I mentioned in the previous review, the authors suggested that there were several interventions that have been shown to be effective, but the references given were not sufficient to prove this, and the criteria for choosing the references were not clear either. The authors have work on its part and propose new references. I took the time to read the articles of the references, but unfortunately, my conclusion is still the same. Here I can give you my arguments:

o   For TF-CBT, the authors provide a new reference, from 2016, with studies from 1980 to 2014, which it’s not very recent. The two references of the first version are always used, but as I mentioned in my first review, they are only case reports, the first with 6 cases, and the second with one case. Both studies are interesting, but not sufficient. Moreover, they are both from the same author. I can accept that CBT is indicated, but the arguments used are poor at my point of view. 

o   For the intervention called "My way" the authors didn’t change nothing. The unique reference is a small pilot study, interesting, but in my opinion not enough to say that it is a "proven intervention". Much more research needs to be done on this method to prove it.

o   For the "Teaching Recovery Techniques", as I mentioned in my first review the reference is from 2018. The study is interesting and well done, and the results are consistent, showing that this intervention can be effective, but the study was not sufficient to confirm this. In this second version of the paper, the authors included a new reference (Saradi et al. 2020), explaining that they started a study to evaluate the effect and efficiency of this approach in a randomized controlled trial. This study it’s the protocol that I mentioned in my previous review, but as I mentioned too, the study was not completed due to recruitment problems and the Covid pandemic : “Then the same group of researchers worked on a very interesting protocol, published in 2020. The aim of this second study was to do a randomised trial, but unfortunately the study was not completed due to recruitment problems and finally the Covid pandemic, as they mention in their 2022 publication. I hope they will continue to advance the research in the future. "Rondung.E, Leiler.A, Sarkadi.A, Bjärtå.A, Lampa.E, Löfving.S, Calam.R, Oppedal.B, Keeshin.B, Warner.G. (2022). Feasibility of a randomised trial of Teaching Recovery Techniques (TRT) with refugee youth: results from a pilot of the Swedish UnaccomPanied yOuth Refugee Trial (SUPpORT). Pilot and feasibility studies. 8. 10.1186/s40814-022-00998-1.”. Then, I can clearly said that this technique it’s not enough proved to be effective. The authors didn’t finish to follow the news results after the starting of the study of 2020. 

o   Concerning the "Expressive Arts Intervention (EXIT)" the authors didn’t change nothing else. The study mentioned was interesting and well done. The intervention can be considered effective in improving the quality of life of unsupported young people.

o   Finally, regarding the "Promotion of acculturation and intercultural competence", there are many empirical studies and some research proving its effectiveness. There were not needed other changes. 

Author Response

Author’s response to Reviewer 1 second review

Brief reviewer’s summary : 

The authors have actualized the introduction with interesting new information, allowing the article to be more actual. Also, they explain why they are limited to only studies written in English. But, unfortunately, the main problem for this paper remains the same. As I mentioned in the previous review, the authors suggested that there were several interventions that have been shown to be effective, but the references given were not sufficient to prove this. The authors have work on its part and propose new references. I took the time to read the articles of the references, but unfortunately, my conclusion is still the same. Further, in the “specific comments to the authors”, I give you the arguments. 

Specific comments to the authors :

In the first review I have suggested some corrections, that all have been considered. 

  • Pag. 2, §1 (now 2§2) : now is more clear. Confirmed
  • Pag. 2, §2 (now 3§1): good reference. Confirmed
  • Pag. 3, §2 (now 3§3) : “Deficit mindset” needs still a little bit more of precision about what means. 

 The phrase deficit mindset has been deleted from the text.

  • Pag. 3,§3 (now 4§1): The scoping review has been enlarged to studies from 2019 to 2022. confirmed
  • Pag.9,§1 (now 10§): the concept of resilience is better defined. confirmed
  • Pag.12 : Concerning the “Effective Interventions”: As I mentioned in the previous review, the authors suggested that there were several interventions that have been shown to be effective, but the references given were not sufficient to prove this, and the criteria for choosing the references were not clear either. The authors have work on its part and propose new references. I took the time to read the articles of the references, but unfortunately, my conclusion is still the same. Here I can give you my arguments:
  • For TF-CBT, the authors provide a new reference, from 2016, with studies from 1980 to 2014, which it’s not very recent. The two references of the first version are always used, but as I mentioned in my first review, they are only case reports, the first with 6 cases, and the second with one case. Both studies are interesting, but not sufficient. Moreover, they are both from the same author. I can accept that CBT is indicated, but the arguments used are poor at my point of view. 
  • The reviewer is correct in their critique and assessment of this section. An older reference by Cary and McMillen (2012) has been added at the beginning of the subsection to document the established effectiveness of TF-CBT in children and adolescents. It has been reformatted with additional text and a more recent systematic review of TF-CBT by Chipalo (2021) in refugee children. It should be stated that with limited mental health services and the difficulty of research with unaccompanied refugee children, that the studies are limited but show progress. In addition, at the end of this subsection, a proviso has been added that further empirical research is necessary to further document its effectiveness and generalizability.
  • For the intervention called "My way" the authors didn’t change nothing. The unique reference is a small pilot study, interesting, but in my opinion not enough to say that it is a "proven intervention". Much more research needs to be done on this method to prove it.
  • The pilot study nature of the initial citation is more clearly stated. In addition, a more recent randomized controlled study from 2019 was added to strengthen the promise of this newer intervention approach.
  •  
  • For the "Teaching Recovery Techniques", as I mentioned in my first review the reference is from 2018. The study is interesting and well done, and the results are consistent, showing that this intervention can be effective, but the study was not sufficient to confirm this. In this second version of the paper, the authors included a new reference (Saradi et al. 2020), explaining that they started a study to evaluate the effect and efficiency of this approach in a randomized controlled trial. This study it’s the protocol that I mentioned in my previous review, but as I mentioned too, the study was not completed due to recruitment problems and the Covid pandemic : “Then the same group of researchers worked on a very interesting protocol, published in 2020. The aim of this second study was to do a randomised trial, but unfortunately the study was not completed due to recruitment problems and finally the Covid pandemic, as they mention in their 2022 publication. I hope they will continue to advance the research in the future. "Rondung.E, Leiler.A, Sarkadi.A, Bjärtå.A, Lampa.E, Löfving.S, Calam.R, Oppedal.B, Keeshin.B, Warner.G. (2022). Feasibility of a randomised trial of Teaching Recovery Techniques (TRT) with refugee youth: results from a pilot of the Swedish UnaccomPanied yOuth Refugee Trial (SUPpORT). Pilot and feasibility studies. 8. 10.1186/s40814-022-00998-1.”. Then, I can clearly said that this technique it’s not enough proved to be effective. The authors didn’t finish to follow the news results after the starting of the study of 2020. 

  • The author has clarified in the text that the initial feasibility RCT required adjustments to the recruitment protocol and the second reference provided by the reviewer has been added to the citations. In addition, a new article from 2023 by Solhaug and colleagues highlights TRT usefulness for “satisfaction with life” for refugee you. Solhaug AK, Roysamb E, and Oppedal B. Changes in life satisfaction among unaccompanied asylum-seeking and refugee minors who participated in teaching recovery techniques (TRT). Child and Adolescent Psychiatriy and Mental Health. 2923; 17(50): 1-13. doi.org/10.1186/s13034-023-00595-x
  •  In addition, another pilot study was referenced and cited that examined the feasibility or TRT with a parenting education program which has shown positive and encouraging results. El-Kani A, Cartwright K, Ang C, Henshaw E, Tanveer M, and Calam R. Testing the Feasibility of Delivering and Evaluating a Child Mental Health Recovery Program Enhanced With Additional Parenting Sessions for Families Displaced by the Syrian Conflict: A Pilot Study. Peace and Conflict: Journal of Peace Psychology. 201; 24(2): 188–200

  • Concerning the "Expressive Arts Intervention (EXIT)" the authors didn’t change nothing else. The study mentioned was interesting and well done. The intervention can be considered effective in improving the quality of life of unsupported young people. confirmed
  •  
  • Finally, regarding the "Promotion of acculturation and intercultural competence", there are many empirical studies and some research proving its effectiveness. There were not needed other changes. 
    • The section on acculturation and intercultural competence was moved from the interventions to the Protective Factors section.
  • Finally, it should be noted that the author(s) have added Narrative Exposure Therapy for Children (KIDNET) as another therapeutic approach with references from 2019 and 2020.

Round 3

Reviewer 1 Report

The authors has worked a lot to improve the paper, specially with the references. The work done is enough now. 
Thank you very much for your job.